# Characterization and Quantitative Comparison of Key Aroma Volatiles in Fresh and 1-Year-Stored Keemun Black Tea Infusions: Insights to Aroma Transformation during Storage

**DOI:** 10.3390/foods11050628

**Published:** 2022-02-22

**Authors:** Meng Tao, Wenli Guo, Wenjun Zhang, Zhengquan Liu

**Affiliations:** 1State Key Laboratory of Tea Plant Biology and Utilization, Anhui Agricultural University, Hefei 230036, China; teatmtm@163.com (M.T.); weixinyi@stu.ahau.edu.cn (W.G.); 20720069@stu.ahau.edu.cn (W.Z.); 2School of Tea & Food Science and Technology, Anhui Agricultural University, Hefei 230036, China

**Keywords:** Keemun black tea, GC-O-MS, storage, aroma volatile

## Abstract

The aroma of Keemun black tea (KBT) changes during storage. We investigated key aroma volatiles of fresh KBT (FKBT) and KBT stored for 1 year. Through gas chromatography–olfactometry–mass spectrometry/aroma extract dilution analysis (GC-O-MS/AEDA), 27 aroma volatiles with a flavor dilution (FD) value ≥16 were quantitated. In odor activity value (OAV) analysis, the two samples had nearly the same key aroma volatiles; (Z)-methyl epijasmonate was the exception. Dimethyl sulfide, 3-methylbutanal, 2-methylpropanal, and linalool had especially high OAVs. Except for β-damascenone, volatiles with OAVs > 1 had higher concentrations in FKBT, which revealed that most key aroma compounds were lost during storage. Sweet, malty, floral, and green/grassy aromas corresponded directly to certain compounds. Lastly, the addition test indicated that the addition of several key aroma volatiles decreasing during storage could enhance the freshness of KBT aroma, which may be a potential to control the aroma style of KBT or other teas in industry.

## 1. Introduction

Keemun black tea (KBT) is a famous black tea produced in Keemun and surrounding regions in China. Its characteristic, complex aroma—known as the Keemun aroma—distinguishes it from other black teas. According to our evaluations, fresh KBTs (FKBT) have a strong sweet aroma with floral notes. After several months of storage at room temperature, a woody aroma forms and the tea smells notably different from fresh KBTs.

Storage has considerable effects on tea aromas. The aromas of most green teas transform negatively during room temperature storage. The freshness of the aromas fades easily and a stale aroma forms after long storage and strongly affects the green tea quality [1]. For an oolong tea such as tieguanyin, several months of room temperature storage can weaken its characteristic lavender aroma but strengthen its grassy aroma. This alteration negatively affects the tea’s quality but the use of vacuum packaging can slow the change. However, for white teas, an herbal aroma forms during storage that affects the tea positively and storage can strengthen the characteristic stale aroma of dark teas. Throughout the long history of tea drinking, both the negative and positive effects of storage on tea aromas have been understood. To date, most researchers have focused on the changes of white or dark tea volatiles during storage [2,3,4]. As far as we know, investigations about the effect of storage on black tea volatiles or odor-active compounds are blank.

In earlier research, we investigated aromatic and volatile changes during 1–20 years of storage and identified volatiles linked with storage duration [5]. However, we do not yet know which volatiles contribute to the Keemun aroma or how the key aroma compounds change during storage. Therefore, we selected FKBT and KBT stored for 1 year (1Y-KBT) with a similar matrix. Using gas chromatography–olfactometry–mass spectrometry/aroma extract dilution analysis (GC-O-MS/AEDA), standard addition methods, and sensory experiments, we attempted to identify and compare the key aroma compounds in these two teas that contribute to their main aroma nuances to comprehend aroma transformation during storage.

## 2. Material and Methods

### 2.1. Samples and Chemicals

FKBT and 1Y-KBT (produced in 2020 and 2019, respectively) were acquired from Anhui Guorun Tea Industrial Co., Ltd. (Chizhou City, Anhui, China). The 1Y-KBT sample was stored in a cabin that could keep a relatively stable storage environment. After their acquisition, the samples were stored in a clean, odorless environment at −20 °C. The distilled water used in the experiments was purchased from Watsons Water Company, Inc. (Guangzhou, China). Chromatographically pure methyl tert-butyl ether (MTBE) and pentane were purchased from Aladdin (Shanghai, China) and redistilled before use; 2-octanol and ethyl decanoate (99.5% and 99% pure) were also obtained from Aladdin. *n*-Alkanes (C5–C40; Sigma-Aldrich, St. Louis, MO, USA) were employed to assist in identifying volatiles by enabling the calculation of retention indices. A solid-phase extraction (SPE) cartridge (LC-Si, 500 mg/6 mL) and a solid-phase microextraction (SPME) needle (50/30 μm DVB/Carboxen/PDMS) were purchased from Supelco (St. Louis, MO, USA). All reference aroma compounds were chromatographically pure and commercially obtained, and the actual purities were confirmed through GC-MS.

### 2.2. Isolation of Tea Volatiles

For the isolation of volatiles, 6 g of each sample was infused in 300 mL of distilled water for 5 min, filtered, and cooled in ice water for 5 min. The infusions were extracted three times with 60 mL of 4:1 pentane:MTBE. This SPE procedure was conducted as previously described [6]. For the SPME procedure, 10 mL of each tea infusion was sealed in a 20 mL vial, equilibrated, and extracted with the SPME needle at 30 °C for 30 min.

### 2.3. Fractionation of Acidic and Neutral-Basic Tea Volatiles

The fractionation of acidic and neutral-basic tea volatiles followed a procedure described in a previous paper [7]. KBT volatiles were isolated and extracted as described in the previous section. Acidic volatiles were isolated by shaking the organic extract with 50 mL of aqueous sodium carbonate solution (0.5 mol/L) three times. A total of 150 mL of aqueous solution was washed with MTBE (35 mL) and acidified to pH 2 by using hydrochloric acid (32%), and the acidic volatiles were re-extracted three times with 50 mL of 4:1 pentane: MTBE. The organic phases were dried over anhydrous sodium sulfate and non-volatiles were removed using SPE as described in 2.2, which were then concentrated to 200 μL by a gentle nitrogen stream (acidic fraction). After being washed by aqueous sodium carbonate solution, the remaining organic phase contained only neutral-basic tea volatiles and the next steps were identical to the previous steps (neutral-basic fraction).

### 2.4. Gas Chromatography–Olfactometry–Mass Spectrometry

Aliquots of samples (2 μL) or SPME needles were injected manually in splitless mode. The effluent of the capillary column was split evenly (1:1) between a sniffing port (200 °C) and an MS detector (MSD). The GC-MS system employed was the Agilent 7890B-5975B (Agilent, Santa Clara, CA, USA) equipped with the Agilent DB-5MS and DB-WAX (30 m × 0.25 mm × 0.25 μm) capillary-type chromatographic columns. The GC-O-MS experiments were conducted primarily using the DB-5MS column and the DB-WAX column was used to assist in identification. The GC settings were as below. For the DB-5MS column, the oven temperature was programmed to increase from 50 °C (5 min hold) to 230 °C (2 min hold) at a rate of 5 °C/min, then to 280 °C (1 min hold) at 15 °C/min, and 50 °C (5 min hold) to 230 °C (30 min hold) at a rate of 5 °C/min for the DB-WAX column. Mass spectrometer conditions were as follows: ionization mode, electron impact (EI); ionization energy, 70 eV; ion source temperature, 230 °C; quadrupole temperature, 150 °C; and quadrupole mass spectrometer scan range, 30–350 atomic mass units (amu).

### 2.5. Aroma Extract Dilution Analysis

Tea volatile concentrates (acidic and neutral-basic fractions) were diluted stepwise with 1:1 (by volume) pentane:MTBE, injected, and analyzed using GC-O-MS as described above. Three trained panelists from our department participated. The flavor dilution (FD) factor of each aroma volatile was defined as the number of the greatest dilution at which its smell could be perceived.

### 2.6. Identification of Aroma Volatiles

The identification of aroma volatiles proceeded as described in our previous paper about KBT. Sensory descriptions of aroma volatiles were also used to assist in identification. In addition, reference chemicals were used to help identify or eliminate severely coeluted compounds.

### 2.7. Descriptive Sensory Analysis of KBT

Descriptive sensory analysis (DSA) was conducted using the tea infusions obtained through the procedure described in Section 2.2. To represent KBT aroma and to train the panelists, the following five aroma nuances and their corresponding chemicals were used: malty (3-methylbutanal), sweet (β-damascenone and coumarin), flowery (linalool), green/grassy (hexanal), and woody (4-oxoisophorone). Eight tea evaluators in our laboratory scored the tea aromas from 0 to 5 (with 0.5 gradation). The results are presented as averages.

### 2.8. Quantitation of Odor-Active Compounds through Standard Addition

Approximately 1 mg/mL stock solutions of odor-active compounds and internal standards (2-octanol and ethyl decanoate) for standard addition were prepared in absolute ethanol and stored at 4 °C. The compounds and internal standard were added to the tea infusions and stirred at 300 r/min for 10 min to equilibrate. The compounds were extracted through SE-SPE or desorbed through SPME according to their volatility. Data were collected using GC-MS in the selected-ion monitoring (SIM) or extracted-ion chromatograph mode (EIC). The qualifier and quantifier ions of the compounds are listed in Table 1. Quantitation was accomplished by building standard curves of the amount of each compound added versus the ratios of the peak areas of the quantifier ions and internal standard. Five points, including a blank, were used to build the standard curves.

### 2.9. Calculation of Odor Activity Values

The odor activity values (OAVs) of the odor-active compounds were calculated by the ratio of concentrations and the thresholds in water of these compounds.

### 2.10. Addition Test

Three glasses of fresh tea infusions were prepared as described above and one of them was added accurate amounts of odor-active compounds to reach same concentrations in the FKBT infusion, expect for β-damascenone. The infusions were incubated at 50 °C for 10 min and presented in a random order to 10–13 assessors who orthonasally smelled them. The assessors were also asked to describe the sensory differences.

## 3. Results and Discussion

### 3.1. DSA Results for FKBT and 1Y-KBT

FKBT and 1Y-KBT with similar matrices were chosen for this study. As Figure 1 indicates, the aroma of the two teas greatly differed. The FKBT was characterized by a highly sweet and floral aroma, and the 1Y-KBT was characterized by a sweet and woody aroma. In addition, the FKBT had stronger malty and green/grassy scents that made it smell fresher. The aromas of the two samples reflected the effect of storage on the transformation of the Keemun aroma; these samples can be used for later experiments.

### 3.2. Aroma Volatiles in FKBT and 1Y-KBT

According to our previous experiments, KBT has a complex volatile profile, with a high concentration of acidic volatiles. To avoid the coelution of volatiles, the acidic and neutral-basic tea volatiles were separated. First, GC-O-MS/AEDA was conducted on samples extracted through SE-SPE; all odor-active compounds with an FD value ≥16 are shown in Table 2A. For compounds that remained coeluted, reference chemicals and aroma descriptions were used to assist with identification, for example, in the confirmation of methional and elimination of dihydroactinidiolide.

Despite differences in FD, all of the investigated odor-active compounds were present in both samples. Compatible with our understanding of the Keemun aroma and with the DSA results, the most frequent aroma descriptor obtained from GC-O was ‘sweet’. The various sweet aroma components included sulfur compounds, lactones, aldehydes, and ketones. Among them, lactones were the most abundant. In studies on odor-active compounds in black tea [8,9,10,11,12,13,14,15], raspberry ketone and herniarin have been reported as the primary sweet compounds; β-damascenone and vanillin, also common sweet compounds in black tea, were detected in this experiment. However, furanone, another common sweet component, was not detected in the current GC-O-MS experiments or in our previous study of KBT.

Compounds with a floral aroma were the second most abundant, including alcohols, aldehydes, acids, and esters. The third-most common compounds were all aldehydes with a cucumber-like aroma; among them, (E,Z)-2,6-nonadienal has frequently been described as smelling cucumber-like in GC-O studies. Unexpectedly, another component with a cucumber-like aroma, neral, was discovered but it has not been reported. The other compounds were described as green, leather, herbal, or orange-peel-like; except for benzoic acid, all such others have been reported as odor-active compounds in black tea as cited above.

Although a woody aroma was characteristic of KBT after storage, no odor-active compounds were described as woody. We previously analyzed aroma volatiles in KBT stored for 3 and 15 years through GC-O-MS and similarly discovered no woody aroma volatiles (data not shown). After 1Y-KBT concentrate was dripped onto a smelling strip, a woody aroma was smelled; therefore, the extraction method and solvent used may be appropriate. In our previous study of the key volatiles related to the storage of KBT, we observed that the concentrations of some volatiles formed through the degradation of lipids and carotenoids increased during storage. We therefore inferred woody aroma volatiles to be among them; however, in GC-O-MS, most of these volatiles did not contribute to the KBT aroma.

Highly volatile compounds cannot be analyzed through methods based on solvent extraction because of the loss during the concentration and cover by the solvent peak. Therefore, a static headspace technique or SPME is generally used to trap the highly volatile compounds and supply GC-O data. In the present study, SPME was employed. Due to the matrix differences between the samples and differences among the volatiles in their degree of absorption, AEDA was not applied in the SPME-GC-O-MS and we recorded only whether aroma volatiles were perceived. Differences in the highly volatile compounds between the two samples were quantitated. As Table 2B indicates, four highly volatile compounds existed in both the FKBT and 1Y-KBT: 2-methylpropanal, 3-methylbutanal, 2-methylbutanal, and dimethyl trisulfide. Dimethyl sulfide was present only in FKBT. In addition, dimethyl disulfide was a common odor-active sulfur compound [16,17,18] and dihydroactinidiolide exhibited a linear increase during storage [5]; thus, these two volatiles were also quantitated.

### 3.3. Quantitative Differences between FKBT and 1Y-KBT

The standard addition method could be used for the accurate quantitation of volatile compounds, which could avoid the matrix effect and mimic losses during extraction and concentration [19]. The SIM mode of GC-MS improved the accuracy of quantification. The concentrations of the aroma volatiles with an FD value ≥16 in both KBT infusions are presented in Table 3A. The regression correlation coefficients (R^2^) of compounds quantified through standard addition ranged from 0.9888 to 0.9999 (benzeneacetic acid was semi-quantified using an internal standard because of the lack of a reference chemical), which was acceptable for precise quantification. Table 3B contains the OAVs of these compounds. Compared with FD, the quantitative analysis more directly indicated the differences between FKBT and 1Y-KBT. The concentrations of aroma volatiles changed dramatically after 1 year of storage; the ratios of the compound concentrations in FKBT to those in 1Y-KBT ranged from 0.3 to 10.9. After OAV calculation, FKBT and 1Y-KBT had 15 and 14 compounds, respectively, with an OAV > 1; all of the compounds, apart from (Z)-methyl epijasmonate were shared between them. Furthermore, all the aroma volatiles with an OAV > 1, apart from β-damascenone, had a higher OAV in FKBT than in 1Y-KBT.

Dimethyl sulfide, 2-methylpropanal, 3-methylbutanal, and 2-methylbutanal, which were detected through SPME-GC-O-MS, were measured at >300 μg/L in the FKBT infusion much higher than in the 1Y-KBT infusion. The ratios of the concentrations of these four volatiles in FKBT to those in 1Y-KBT were between 4.0 and 10.9. After OAV calibration, the OAVs of all four of these volatiles exceeded 100 in the FKBT infusion. In particular, dimethyl sulfide had an OAV > 1000. Although the concentrations of these volatiles were much lower in the 1Y-KBT infusion, they contributed greatly to the aroma. These four compounds are commonly detected in various teas and products that undergo the Maillard reaction and have clear metabolic pathways [20]. Some evidence indicates that these compounds exhibit an increasing trend with infusion time [8,16].

Except for dimethyl sulfide, the concentrations of the sulfide compounds (dimethyl disulfide, methional, and dimethyl trisulfide) were also higher in the FKBT infusion than in the 1Y-KBT infusion; the ratios were 7.9, 4.1, and 2.4, respectively. In an earlier analysis of KBT after 1 to 20 years of storage, dimethyl disulfide was detected in only 1Y-KBT and neither methional nor dimethyl trisulfide were detected in any stored samples. Methional is a Strecker aldehyde of methionine and both dimethyl disulfide and dimethyl trisulfide are further oxidation products of methional [20]. Methional’s aroma has been described as cooked potato in several reports [21,22] but we linked it with the characteristic sweet aroma of high-grade FKBT in our GC-O-MS experiment. The concentration of methional in the FKBT infusion was 4.2 μg/L and those of both methyl disulfide and dimethyl trisulfide were <1.0 μg/L. The OAVs of methional and dimethyl trisulfide were >1 in both the FKBT and 1Y-KBT infusions, indicating that these compounds contributed to the aromas.

Among the six volatiles with a floral aroma, benzeneacetaldehyde was slightly more abundant in the FKBT infusion and the concentrations of linalool, geraniol, and (Z)-methyl epijasmonate were 2.5, 2.2, and 5.1 times higher in the FKBT infusion; except for that of (Z)-methyl epijasmonate, the concentrations of the compounds were all >100 μg/L. The concentration of phenylethyl alcohol was slightly lower in the FKBT infusion than in the 1Y-KBT infusion but both concentrations were >400 μg/L. These results generally support our previous conclusions that storage has a negative effect on floral compounds, especially linalool, geraniol, and (Z)-methyl epijasmonate. Linalool, geraniol, and benzeneacetaldehyde contributed greatly to the floral aroma of the FKBT infusion, especially linalool, which had an OAV of 257. Although it was considerably reduced during storage, the OAV of linalool remained at 102 in the 1Y-KBT infusion. (Z)-methyl epijasmonate, a key aroma volatile in tea, contributed only slightly to the aroma of the FKBT infusion, with an OAV of 1. The acidic aroma volatiles exhibited similar trends. The concentrations of benzeneacetic acid and benzoic acid in the 1Y-KBT infusion were 4.4 and 3.1 times higher, respectively, than in the FKBT infusion.

Hexanal, the only green/grassy aroma volatile with an FD value ≥16, is derived from lipid oxidation during tea processing [20]. The concentration of this compound in the FKBT infusion (165 μg/L) was 4.2-folds higher than in the 1Y-KBT infusion. The OAVs of hexanal in both KBT infusions were higher than 1.

The concentrations of all eight sweet aroma volatiles in both KBT infusions were <30 μg/L. Methional, coumarin, and δ-decalactone were more abundant in the FKBT infusion and the other five sweet volatiles were more abundant in the 1Y-KBT infusion. The concentrations of δ-octalactone, vanillin, and herniarin in the FKBT infusion were all 0.9 times lower than those in the 1Y-KBT infusion and the concentrations of β-damascenone and raspberry ketone were 0.6 and 0.3 times lower than those in the 1Y-KBT infusion. Among the sweet aroma volatiles, only β-damascenone, methional, and coumarin had OAVs > 1. Though low concentrations of β-damascenone were present in the KBT infusions (both <1 μg/L), its OAVs were still much higher (61 and 108 in FKBT and 1Y-KBT, respectively) than those of methional and coumarin because of its extremely low threshold.

β-Damascenone receives tremendous attention because it can easily achieve high FD in GC-O experiments and has a high OAV due to its extremely low threshold. β-damascenone exists in various teas, fruits, coffees, honeys, wines, and other foods and drinks [23]. The sensory effect of β-damascenone in wine is especially concentrated. On the basis of GC-O results and OAVs, many researchers have concluded that β-damascenone contributes greatly to wine’s aroma. However, sensory tests have indicated that the threshold of β-damascenone is easily affected by the matrix. Its thresholds in water and wine are 0.004 and 7 µg/L, respectively, but its concentration in wine is 1–1.5 µg/L; thus, β-damascenone does not affect wine’s aroma directly and may contribute only indirectly to the fruity aroma of wine [24]. Compared with studies on the sensory effect of β-damascenone in wine, those on its effect in tea aroma are few. Research studies about the β-damascenone sensory effect indicated to us that checking the actual sensory effect of aroma volatiles was necessary.

The concentrations of two cucumber-like aroma volatiles, namely (E,Z)-2,6-nonadienal and neral, in the FKBT infusion were 1.9-folds higher than those in the 1Y-KBT infusion, but still <5 μg/L. The OAVs of (E,Z)-2,6-nonadienal were 49 and 26 in the FKBT and 1Y-KBT infusions, respectively, but neral made no contribution to either KBT infusion aroma, with an OAV <0.1. In addition, (E)-2-nonenal, frequently detected in various teas through GC-O, contributed slightly to both KBT aromas, with OAVs of 2 and 1 in the FKBT and 1Y-KBT infusions, respectively. Although dihydroactinidiolide was reported to be an aroma volatile in several teas [3,25,26], we did not perceive its aroma. However, the concentration of dihydroactinidiolide in the FKBT infusion was only 60% of that in the 1Y-KBT infusion according to standard addition, supporting our previous conclusion that dihydroactinidiolide was a reliable indicator of storage duration.

### 3.4. Additional Test to Understand Aroma Transformation during Storage

Additional tests were designed to further investigate the transformation of aroma during KBT storage. Considering all aroma volatiles with OAVs > 1, apart from β-damascenone, were more abundant in the FKBT than in the 1Y-KBT infusion, we deduced that the loss of key aroma volatiles during storage may contribute to the transformation of KBT aroma. Therefore, to evaluate the effects of individual compounds on the aroma, aroma compounds with an OAV of >1 in the 1Y-KBT infusion were added to match the levels of those in the FKBT infusion and β-damascenone was added to 150% of the level in the 1Y-KBT infusion for triangle testing to determine significant differences.

As illustrated in Table 4, several aroma volatiles significantly inhibited the woody aroma, especially 3-methylbutanal, dimethyl sulfide, and benzeneacetaldehyde; (E,Z)-2,6-nonadienal, 3-methylbutanal, dimethyl sulfide, and (E,Z)-2,6-nonadienal also made the infusion feel fresher. In addition, the addition of benzeneacetaldehyde, linalool, and geraniol increased the strength of the floral aroma of the infusion and both β-damascenone and methional made the infusion aroma sweeter. The sensory evaluation results indicated a direct contribution of these compounds to the KBT aroma.

## 4. Conclusions

GC-O-MS/AEDA and quantitative experiments with FKBT and 1Y-KBT demonstrated that approximately two-thirds of the aroma volatiles were more abundant in FKBT; these included high-volatility compounds, sulfide compounds, and some floral volatiles. Approximately one-third of the aroma volatiles were more abundant in 1Y-KBT, including benzeneacetic acid, benzoic acid, and raspberry ketone. However, most compounds with an OAV > 1 were more abundant in the FKBT infusion, with the exception of β-damascenone. In addition, we understood the transformation of KBT aroma through additional testing, in which we observed that the additions of several key aroma volatiles (including 3-methylbutanal, dimethyl sulfide, benzeneacetaldehyde, (E,Z)-2,6-nonadienal, and so on) weakened the woody aroma and enhanced the freshness of the KBT aroma, which indicated that controlling the storage environment to prevent the loss of these key aroma volatiles was very important to make KBT fresh.

## Figures and Tables

**Figure 1 foods-11-00628-f001:**
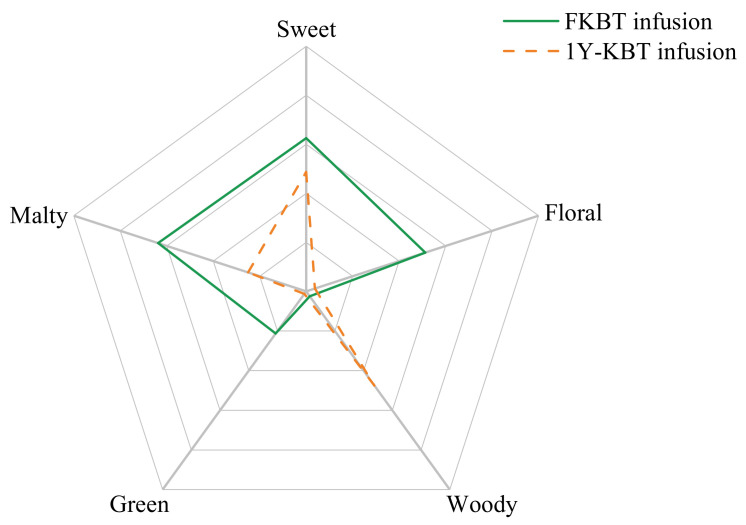
Descriptive sensory analysis results of FKBT and 1Y-KBT.

**Table 1 foods-11-00628-t001:** Qualifier and quantifier ions of key aroma volatiles.

Name	CAS	Quantifier Ion	Qualifier Ions
Dimethyl sulfide	75-18-3	62	62; 47; 61
2-Methyl-propanal	78-84-2	72	43; 41; 72
3-Methyl-Butanal	590-86-3	44	44; 41; 43
2-Methyl-Butanal	96-17-3	57	57; 58; 41
Dimethyl disulfide	624-92-0	94	94; 79; 45
Hexanal	66-25-1	56	56; 44; 41
Methional	3268-49-3	48	48; 104; 47
Dimethyl trisulfide	3658-80-8	126	126; 79; 45
Benzeneacetaldehyde	122-78-1	91	91; 120; 92
Linalool	78-70-6	71	71; 93; 55
Phenylethyl alcohol	60-12-8	91	91; 92; 122
(E,Z)-2,6-Nonadienal	557-48-2	69	41; 70; 69
(E)-2-Nonenal	18829-56-6	70	70; 43; 55
Benzoic acid	65-85-0	105	105; 122; 77
(Z)-Linalool oxide (pyranoid)	14009-71-3	68	68; 94; 59
Neral	106-26-3	69	69; 41; 39
Geraniol	106-24-1	69	69; 41; 68
δ-Octalactone	698-76-0	99	99; 71; 42
Eugenol	97-53-0	164	164; 103; 149
β-Damascenone	23726-93-4	69	69; 121; 190
Vanillin	121-33-5	151	151; 152; 81
Coumarin	91-64-5	146	146; 118; 90
δ-Decalactone	705-86-2	70	99; 71; 70
Dihydroactinidiolide	15356-74-8	111	111; 109; 43
Raspberry ketone	5471-51-2	107	107; 43; 164
(Z)-Methyl epijasmonate	39924-52-2	224	83; 151; 224
Herniarin	531-59-9	176	176; 148; 133

**Table 2 foods-11-00628-t002:** (**A**) SPE-GC-O-MS/AEDA results of FKBT and 1Y-KBT. (**B**) SPME-GC-O-MS results of FKBT and 1Y-KBT.

(A)								
Name ^a^	CAS	RI		Description	Fraction ^b^		FD ^c^	
		DB-5MS	DB-WAX				FKBT	1Y-KBT
Hexanal	66-25-1	800	1076	Green	NBF, AF		16	8
Methional	3268-49-3	902	1459	Sweet	NBF		16	2
Benzeneacetaldehyde	122-78-1	1042	1649	Floral	NBF		64	64
Linalool	78-70-6	1100	1551	Floral	NBF		128	128
Phenylethyl alcohol	60-12-8	1110	1918	Floral	NBF, AF		256	256
(E, Z)-2,6-Nonadienal	557-48-2	1152	1588	Cucumber	NBF		16	16
(E)-2-Nonenal	18829-56-6	1157	1165	Leather	NBF, AF		32	32
Benzoic acid	65-85-0	1165	2443	Leather	NBF, AF		8	32
(Z)-Linalool oxide (pyranoid)	14009-71-3	1174	1766	Herbal	NBF		8	32
Benzeneacetic acid	103-82-2	1244	2570	Floral	AF		8	32
Neral	106-26-3	1237	1684	Cucumber	NBF		64	64
Geraniol	106-24-1	1249	1851	Floral	NBF		512	128
δ-Octalactone	698-76-0	1278	1973	Sweet	NBF		16	8
Eugenol	97-53-0	1349	2177	Dried orange peel	NBF		64	256
β-Damascenone	23726-93-4	1379	1378	Sweet	NBF, AF		512	1024
Vanillin	121-33-5	1393	2578	Sweet	AF		128	64
Coumarin	91-64-5	1435	2467	Sweet	NBF, AF		256	256
δ-Decalactone	705-86-2	1491	2234	Coconut	NBF		64	64
Raspberry ketone	5471-51-2	1548	3003	Sweet	AF		256	256
(Z)-Methyl epijasmonate	39924-52-2	1672	2354	Floral	NBF		64	32
Herniarin	531-59-9	1724	2948	Sweet	NBF, AF		128	128
**(B)**								
**Name ^a^**	**CAS**	**RI (DB-5)**		**Description**	**FKBT ^d^**	**1Y-KBT**		
Dimethyl sulfide	75-18-3	518		Cooked corn	√	×		
2-Methyl-propanal	78-84-2	551		Malty	√	√		
3-Methyl-butanal	590-86-3	647		Malty	√	√		
2-Methyl-butanal	96-17-3	657		Malty	√	√		
Dimethyl trisulfide	3658-80-8	965		Stinking	√	√		

^a^ Aroma volatiles were identified by mass spectrometry, RI index, and GC-O data. For low abundance or co-elution compounds, reference chemicals were used. Compounds were ordered with their elution order in DB-5 column. ^b^ NBF, neutral and basic fraction; AF, acid fraction. ^c^ Aroma volatiles with FD ≥ 16 in at least one sample were presented. FKBT, fresh keemun black tea; 1Y-KBT, 1-year keemun black tea. ^d^ ‘√’ indicates compounds detected as aroma volatiles; ‘×’, contrarily.

**Table 3 foods-11-00628-t003:** (**A**) Concentrations of key aroma volatiles in FKBT and 1Y-KBT infusions. (**B**) OAVs of key aroma volatiles in FKBT and 1Y-KBT infusions.

(A)					
Name	CAS	Concentration (μg/L)			Ratios (FKBT:1Y-KBT)
		FKBT	1Y-KBT		
Dimethyl sulfide	75-18-3	379	37		10.4
2-Methyl-propanal	78-84-2	584	145		4.0
3-Methyl-Butanal	590-86-3	715	65		10.9
2-Methyl-Butanal	96-17-3	824	108		7.6
Dimethyl disulfide	624-92-0	0.87	0.11		7.9
Hexanal	66-25-1	135	32		4.2
Methional	3268-49-3	4.2	1.0		4.1
Dimethyl trisulfide	3658-80-8	0.14	0.06		2.4
Benzeneacetaldehyde	122-78-1	511	445		1.1
Linalool	78-70-6	149	59		2.5
Phenylethyl alcohol	60-12-8	416	519		0.8
(E,Z)-2,6-nonadienal	557-48-2	1.5	0.8		1.9
(E)-2-Nonenal	18829-56-6	0.70	0.37		1.9
Benzoic acid	65-85-0	9.9	31		0.3
(Z)-Linalool oxide (pyranoid)	14009-71-3	620	344		1.8
Benzeneacetic acid	103-82-2	4.8	21		0.2
Neral	106-26-3	4.2	2.2		1.9
Geraniol	106-24-1	311	138		2.2
δ-Octalactone	698-76-0	13	13		0.9
Eugenol	97-53-0	0.52	0.73		0.7
β-Damascenone	23726-93-4	0.25	0.43		0.6
Vanillin	121-33-5	10	12		0.9
Coumarin	91-64-5	28	17		1.7
δ-Decalactone	705-86-2	4.5	2.2		2.0
Dihydroactinidiolide	15356-74-8	23	40		0.6
Raspberry ketone	5471-51-2	0.29	0.90		0.3
(Z)-Methyl epijasmonate	39924-52-2	2.8	0.56		5.1
Herniarin	531-59-9	19	21		0.9
**(B)**					
**Name**	**CAS**	**OAV**			
		**FKBT**		**1Y-KBT**	
Dimethyl sulfide	75-18-3	1264		122	
3-Methyl-Butanal	590-86-3	595		54	
2-Methyl-propanal	78-84-2	307		76	
Linalool	78-70-6	257		102	
2-Methyl-Butanal	96-17-3	187		25	
Geraniol	106-24-1	97		43	
Benzeneacetaldehyde	122-78-1	81		71	
β-Damascenone	23726-93-4	61		108	
(E,Z)-2,6-Nonadienal	557-48-2	49		26	
Dimethyl trisulfide	3658-80-8	14		6	
Hexanal	66-25-1	14		3	
Methional	3268-49-3	10		2	
Coumarin	91-64-5	3		2	
(E)-2-Nonenal	18829-56-6	2		1	
(Z)-Methyl epijasmonate	39924-52-2	1		<1	
Dimethyl disulfide	624-92-0	<1		<0.1	
Phenylethyl alcohol	60-12-8	<1		<1	
Vanillin	121-33-5	<1		<1	
Eugenol	97-53-0	<1		<1	
δ-Decalactone	705-86-2	<1		<0.1	
δ-Octalactone	698-76-0	<1		<1	
(Z)-Linalool oxide (pyranoid)	14009-71-3	<1		<0.1	
Benzoic acid	65-85-0	<1		<1	
Dihydroactinidiolide	15356-74-8	<0.1		<0.1	
Neral	106-26-3	<0.1		<0.1	
Raspberry ketone	5471-51-2	<0.1		<0.1	
Herniarin	531-59-9	<0.1		<0.1	
Benzeneacetic acid	103-82-2	<0.1		<0.1	

**Table 4 foods-11-00628-t004:** Addition tests’ results to explain woody aroma.

Added Compounds ^a^	Correct Answers/Panelists ^b^	*p* Value ^c^	Significant Level ^d^	Difference Description
3-Methyl butanal	11/12	0.000	***	More fresh, less woody
Dimethyl sulfide	10/12	0.001	**	More fresh, less woody
Benzeneacetaldehyde	9/11	0.001	**	More floral, less woody
(E,Z)-2,6-Nonadienal	9/11	0.001	**	More fresh, less woody
(E)-2-Nonenal	7/10	0.020	*	More fatty, less woody
Linalool	7/11	0.039	*	More floral, less woody
β-Damascenone (+50%)	7/11	0.039	*	Sweeter, less woody
Methional	7/11	0.039	*	Sweeter, less woody
Geraniol	7/11	0.039	*	More floral, less woody
2-Methyl-propanal	6/12	0.178	-	-
Hexanal	5/10	0.213	-	-
Coumarin	5/10	0.213	-	-
2-Methyl butanal	4/10	0.441	-	-
Dimethyl trisulfide	4/10	0.441	-	-

^a^ Aroma volatiles were added in 1Y-KBT infusion individually, same as the concentrations in FKBT infusion, and β-damascenone was added to150% level than original. ^b^ Number of correct answers/all addition test participants. ^c^ The statistical significance (*p* value) of the addition test was evaluated by binomial test tables. ^d^ ***, very highly significant (*p* < 0.001); **, highly significant (0.001 ≤ *p* ≤ 0.01); *, significant (0.01< *p* ≤ 0.05); and -, not significant (*p* > 0.05).

## Data Availability

Data is contained within the article.

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
