# Peer review of "Characterization and Quantitative Comparison of Key Aroma Volatiles in Fresh and 1-Year-Stored Keemun Black Tea Infusions: Insights to Aroma Transformation during Storage"

_foods, 2022, doi:10.3390/foods11050628_

Round 1

Reviewer 1 Report

The manuscript is written well but experiments are not explained properly. 

1) Did the Authors mention the storage condition ? 2) What is the normal storage condition for most of the teas in China?  3) Authors did not mention the other flavonoids which might be the most important components for characterizing the tea.  4) What library authors use to characterize the aromatic volatile compounds. 

 This paper describes about characterization and quantitative comparison of key aroma volatiles in fresh and 1-year-stored Keemun black tea infusions by using gas chromatography-olfactometry-mass spectrometry/aroma extract dilution analysis(GC-O-MS/AEDA). This study also scientifically revealed the previously unknown effect of storage. In addition, the characteristics of the key aroma volatiles of Keemun black tea can be chemically clarified, which greatly contributes to the improvement of value. The results of additional tests show that adjusting the storage environment is very important for making fresh KBT in order to prevent the loss of key aroma volatiles, which is a hint for the development of the optimum processing method. The content is suitable for this journal, and we evaluate it as valuable as a dissertation.    

 I think it would be a better paper if there was a consideration of the aroma transformation mechanism during storage.

Reviewer 2 Report

This paper describes about characterization and quantitative comparison of key aroma volatiles in fresh and 1-year-stored Keemun black tea infusions.The content is suitable for this journal, and we evaluate it as valuable as a dissertation.

I think it would be a better paper if there was a consideration of the aroma transformation mechanism during storage.

Reviewer 3 Report

Foods

Characterization and quantitative comparison of key aroma volatiles in fresh and 1-year-stored Keemun black tea infusions: insights to aroma transformation during storage

Dear Editor,

In this study, characterization and quantitative comparison of key aroma volatiles in fresh and 1-year-stored Keemun black tea infusions were determined. In general, the manuscript has been well designed and written. However, more information should be given for the analyses done in the research.  My specific comments and questions are below;

  • Line 40: Give literature for this sentence!
  • Line 91: Give more information about the modifications and GC conditions!
  • Line 96: Are three panelists enough for that process?
  • Line 112: What were used as the internal standards?
  • Lines 128 and 129: Give some information about the incubation conditions (time etc.)!
  • There is no information about the statistical analysis applied in the research!
  • The authors should give some information about method validation parameters!

Round 2

Reviewer 1 Report

Accept in the present form.